# Depression in the COVID-19 endemic era: Analysis of online self-disclosures by young South Koreans

**Seoyoung Kim**[1], **TaeYoon Aum**[2], **Dong-gwi Lee**[2] *

1 Yonsei Psychological Science Innovation Institute, Yonsei University, Seoul, Republic of Korea,
2 Department of Psychology, Yonsei University, Seoul, Republic of Korea

* lee82@yonsei.ac.kr

**Data Availability Statement:** Our raw data (the autobiographical online posts about pandemic and endemic blues, N = 1740) is now publicly available on the first author Seoyoung Kim's Github

## Abstract

Although COVID-19 has been declared endemic in South Korea, there are economic and psychosocial after-effects. One of these is the prevalence of depression. Depressed adolescents and young adults struggle with insecurity, loneliness, and lack of confidence due to the life limitations imposed during the pandemic. Young South Koreans experienced deterioration in mental health because of the recurrence of mass infections. To address professionals' concerns about the lingering effects of COVID-19 on youth mental health, we text-mined young South Koreans' online posts about depression during the pandemic and the endemic phases—from February 2020 to May 2023. We used a total of 1,740 selected posts (raw data publicly available on https://github.com/kimalexis1129/PLOS_endemic_depression.git) to explore the situational triggers, additional factors, and by-products of depression that have persisted during the endemic era. We used Latent Dirichlet allocation and Dirichlet-multinomial regression topic modeling methods in conjunction with sentiment analysis and mean comparison. The results showed that the pandemic and endemic topic models shared similarities, but emerging topics showed extended adversities such as adolescents' vulnerability to eating disorders and young adults' tendency to self-isolate. Comparisons between the levels of positive and negative affect during the pandemic and endemic eras revealed no significant changes in mood. We discussed the results in comparison with SARS and MERS precedents and from general and cultural perspectives.

## Introduction

Globally, suicidal behavior has increased since the outbreak of COVID-19 in December 2019 [1, 2]. Compared to pre-COVID-19 rates (3.9–9.2%), Farooq et al. [2] reported that the pooled prevalence of suicidal ideation in the general population during the pandemic was 12.1%. The prolonged epidemic and control measures, such as lockdown and quarantine, have put the general public at risk [3]. People experienced fear of infection, uncertainty, alienation, and loneliness [3, 4]. People with pre-existing mental disorders experienced worsening symptoms [5], and people with no prior history of mental health problems developed psychological

repository: https://github.com/kimalexis1129/PLOS_endemic_depression.git.

**Funding:** The author(s) received no specific funding for this work.

**Competing interests:** The authors have declared that no competing interests exist.

symptoms over time [6]. The most common responses to the pandemic were mood disorders [7], alcohol and substance abuse [8], and posttraumatic symptoms [9]. Across countries and cultures, COVID-19-related depression, known as *the Corona blues* [10], was the most common [11]. Depression is a strong predictor of death by suicide, explaining 13.3% of the variance in suicidality [12], which has increased since COVID-19 [13].

Mental disorders in the general population are concerning because they may persist in the form of a post-epidemic sequelae [14, 15]. Previous experiences with Severe Acute Respiratory Syndrome (SARS) in 2004 and the Middle East Respiratory Syndrome (MERS) in 2015 have shown that distress can persist for years after the end of an outbreak. SARS-related depressive symptoms persisted for three years, and those who underwent quarantine experienced more severe symptoms along with post-traumatic reactions [16, 17]. Similarly, MERS-related mood problems persisted for four to six months after the release from quarantine [18]. Participants experienced emotional exhaustion, numbness, and sleep disturbances [3]. Researchers expected to observe spontaneous mental recovery shortly after the end of the epidemic; however, the study results (e.g., [19, 20]) showed no statistically significant changes in the levels of psychological distress—including depression—despite the return to "normal" circumstances. The SARS and MERS outbreaks lasted only for four and one month, respectively. In comparison, the COVID-19 pandemic lasted for more than three years and resulted in 775,132,086 infections and 7,042,222 deaths worldwide [21]. The follow-up findings after the SARS and MERS suggest that the psychological impact of COVID-19 may be more lasting and profound than that of the previous ones.

Mental health research has been extensive in China and neighboring Asian countries, where high COVID 19 prevalence has made residents vulnerable to corona blues [22, 23]. South Korea gained attention for successfully implementing early social distancing measures; however, the subsequent mass outbreak dampened hopes for disease control [24]. Adolescents and college-aged young adults expressed deep concerns about their education and career development [25]. In an academically competitive and collectivistic culture, young South Koreans faced uncertainty and reduced self-efficacy [26]. Since COVID-19, there has been an immediate increase in the number of suicides among South Korean adolescents and adults [27]. In just two years, adolescent suicide deaths have increased by 20.3%, from 5.9 deaths per 10,000 population in 2019 to 7.1 in 2021 [28]. Relatedly, the number of adolescents and young adults reporting depressive symptoms increased significantly in 2021 compared with 2019 [29]. South Korea had the highest national prevalence of depression among fifteen OECD countries (36.8%), followed by the United States (23.5%) and Japan (17.3%) [30]. Recognizing the severity of the situation, the government provided free psychological counseling and a 24-hour hotline services [31]. However, 62.5% of those who had received psychological services reported persistent depressive symptoms [32].

In May 2023, South Korea declared the end of the COVID-19 pandemic [33]. The government lifted restrictions to restore and normalize people's lives. Schools resumed classes on campus, hoping to address students' concerns about the quality of their education and the lack of career development opportunities. However, as we have learned from the SARS and MERS outbreaks, there are signs of the lingering effects of the corona blues. The reopening of schools helped reduce problems with physical inactivity among adolescents; however, there was an increase in perceived depression, including feelings of loneliness [34]. One reason for this was the stress of peer relationships [35]. Young college graduates suffered from psychosocial consequences related to the economic recession and lack of pre-work experiences, such as internships [36]. This discouraged employment and exacerbated young South Koreans' feelings of insecurity and the lack of competence in general and in work relationships making them more vulnerable to depression related to COVID-19 [37, 38].

American best-selling author Mark Manson noted during his visit in 2024 that South Korea was "the most depressed country" in the world [39]. Indeed, the lifetime prevalence of mental disorders among South Koreans was 27.6% [40], and 6.1% of a representative adult sample ($N$ = 10,710) had clinically relevant depression [41]. The condition has worsened most among young people in their twenties, to the point where they were ranked the lowest in terms of perceived happiness, compared to other age groups [42]. Lee [43] explained that due to situational obstacles such as COVID-19 the young population involuntarily gave up the search for happiness and became depressed. Some cultural characteristics such as high perfectionism and competitiveness [44], and gender and age group conflicts [45, 46] contribute to these struggles, in a vertical collectivist culture [47]. In addition, South Koreans' lower levels of mental health literacy (i.e., recognition of and attitudes of/toward mental disorders, and knowledge and beliefs about treatment) compared to Western countries exacerbate depression [48, 49].

Recent studies on the mental health problems of South Koreans have coined the term *endemic blues* ([50], p. 43). Specifically, online data-driven results by Kim [50] showed that the main keywords were "depression-anxiety," "depression-recovery," and "COVID-aftereffect" from an analysis of 6,554 online posts and comments extracted from various social media (e.g., Twitter/X, YouTube). Similar studies using the online text-mining approach [51–53], have described the severity of the corona blues and the need for long-term psychological support services by the public. For example, Yoo and Lim [53] text-mined 812 South Korean online news articles about public reactions to COVID-19, published from December 2019 to October 2020 and found that "blues", "emotion," and "anxiety" were focal keywords that appeared alongside "COVID" in the news. In the discussions of their findings, the authors emphasized the need for the administration of new mental health control measures to prevent a decrease in the public's immunity due to prolonged exposure to stress and negative emotions. The diversification of online spaces and the accumulation of online discussions have provided a useful source of online text data for researchers—especially during the pandemic when traditional face-to-face research was discouraged [54]. Described by Feldman and Dagan [55] as the "knowledge discovery from text" (p. 112), text-mining from online spaces can provide a phenomenological summary of the public's experiences and opinions [56]. While there has been online-based studies conducted during the pandemic, the ongoing phenomenon of the corona blues calls for a study that compares the presence and content of the endemic blues with the pandemic blues.

Given the high prevalence of suicidality and depression, this study explores the state of the endemic blues among young South Koreans. We used an online text-mining approach to answer the following research questions: (a) Is the residual influence of COVID-19 present in a group of depression-related online posts shared after the pandemic? (b) What are the main themes (causes and contents) of the endemic blues? (c) How have the themes changed over time since COVID-19 was declared as endemic? (d) Has there been a statistically significant decrease in levels of negative affect during the endemic period compared to the pandemic period? As a part of the exploration of themes of struggles with the endemic blues (research question b), we considered age-specific differences between the adolescent and young adult populations.

## Methods

To text-mine depression-related online posts by young South Koreans, we examined publicly available posts on the Knowledge-In forum of the Naver portal (https://kin.naver.com/) using the search term "depression." To compare findings from the posts shared from May 2023 to the end of the year (hereafter, endemic data) with those found by Kim [51] from the posts

from February 2020 to April 2023 (hereafter, pandemic data), the pandemic data ($N$ = 1,132) consisted of the titles and bodies of the knowledge-in post, and the posting dates were obtained from Bum-Hyun Kim. This study does not require Institutional Review Board approval because it does not involve human subjects, but only uses anonymous text data voluntarily made publicly available online. All the results were translated into English by the first author of this study with the help of a professional editor.

### Endemic data collection

On January 14 2024, we scraped the endemic data in the same way as Bum-Hyun Kim [51], who provided us with the pandemic data for comparison, using a web-crawler coded by the first author of this study in a Python 3.6.5 environment using the Selenium framework (https://www.selenium.dev/) and the Windows Firefox web driver. The web-crawler can navigate the Knowledge-In forum in a pre-designed order, from searching with the keyword "depression" to filtering and storing eligible web elements (i.e., post titles, bodies, and posted dates). We saved 10,887 posts with titles containing at least one of the keywords "depress (ion)," "depressed," "depressi(on/ve) symptoms," and "major depressive disorder" from the web crawling (after excluding identical posts shared on the same day as duplicates). In addition, for comparison with the pandemic data, we applied the additional filter keywords "depression" and "diagnos(is/ed)" to the post bodies as done by Kim [51], resulting in the retention of 608 posts.

### Data preparation

Text analysis requires preprocessing of the raw data [57]. Preprocessing includes sentence-to-word tokenization, tagging parts of speech, and extracting meaningful parts (tokens) such as nouns or adjectives that are suitable for specific analysis methods. We used the Korean morpheme-based analyzers KOMORAN (https://www. shineware.co.kr/products/komoran/) and Mecab (https://github.com/koshort/pyeunjeon) together in the Java 17.0.2 and Python 3.6.5 environments to tokenize and tag the pandemic and endemic data. This involved removing meaningless tokens—such as pre/post dispositions as stop words—from sentences and storing meaningful noun and adjective tokens.

In addition, to account for potential age-specific differences in the experience of the endemic blues, the first and second authors of this study manually coded endemic data ($N$ = 608) entries as either "adult," "adolescent" or "unknown" for a subgroup analysis. First, age-relevant keyword filters, such as "middle-" or "high-" "school", "college", "college entrance examinee" and "employment", were applied to post titles to identify those posts that contained some indication of the age range of the anonymous authors ($n$ = 369). Second, the second author scrutinized the post bodies to ensure that the demographic clues were in the descriptions of the writers who were experiencing the endemic blues (and not of surrounding others, such as friends or family members). Lastly, the first author double-checked the coded results and resolved confusion in a few cases ($n$ = 12), such as when a writer referred to a past experience in adolescence, but contextual clues supported that the writer was an adult at the time of writing the post (coded as "adult" in this case).

### Text-analyses

**Topic modeling.** To find the endemic blues-related topics embedded in the data and to investigate how the topics have changed since the declaration of endemicity, we performed two types of topic modeling analyses—Latent Dirichlet allocation (LDA) and Dirichlet-multinomial regression (DMR) methods—in the Java 17.0.2 environment using the MALLET

(Machine Learning for Language E toolkit, https://mimno.github.io/Mallet/) package on March 26 2024. LDA reveals latent topics from observable tokens based on Bayesian inference and random token sampling [58, 59]. Assuming that a corpus of narratives about a phenomenon contains ubiquitous but unknown topics that may represent the public's focal opinions and experiences with respect to the phenomenon, LDA extracts topics that can be inferred from the token- and document-topic (in our case, *an online* post-topic) distributions of probabilities (see [58, 59] for a detailed statistical explanation).

We generated a topic model in the form of a set of high-frequency tokens for each topic, based on an examination of probabilities. An efficient topic model has easily distinguishable differences between topics, and the tokens belonging to a topic have thematic similarities (see a recent example in [60]). For example, a topic model about unemployment issues may include the topic of economic hardship, which consists of tokens such as "debt" and "mortgage," and the topic of psychological struggles, which consists of tokens such as "uncertainty" and "anxiety." Deciding on the number of topics requires consideration of the theoretical interpretability of the model using a perplexity score (the lower the better, see the example in [61, 62]); however, the number of topics in this study was set at six for comparison with the pandemic blues topic model presented in Kim [51]. We then conducted an LDA on the adolescent and adult subgroups of the endemic data to further explore possible age differences. In the subgroup LDA analyses, we extracted five topics for each group, considering the interpretability of the topic models and the size of the data.

We continued with the DMR topic modeling of endemic data. The DMR extends the LDA by adding a third variable—metadata—to the calculation of the probabilistic distributions [63]. Numeric or numerically indexed metadata, such as the year of publication, race, and ethnicity of the author of the document, or location of the publisher, can be used. In this study, we used the month of publication (May to December 2023) to derive the proportional changes in the six topics over time. The topic proportion estimates show how many references were made to a topic relative to other topics and indicate the importance or popularity of a topic [64]. Using DMR, we derived topic proportion estimates by month and visualized the trends using a line graph.

**Sentiment analysis.** We conducted sentiment analysis in the Python 3.6.5 environment using the pyEunjeon interface to compare the levels of negative affect expressed in the pandemic and endemic data, (https://github.com/koshort/pyeunjeon) and the KNU Korean sentiment word dictionary (http://dilab.kunsan.ac.kr/knusl.html). KNU is a lexicon developed by Park et al. [65] based on the standard dictionary of the National Institute of the Korean Language, and consists of 14,843 positive, negative, and neutral sentiment words with pre-assigned sentiment degree scores. A smaller negative score indicates more negative sentiment in a document, a score of zero indicates neutral sentiment, and a larger positive score indicates more positive sentiment.

When we operated the pyEunjeon, the module automatically searched post-by-post and calculated sentiment scores when it found a matching sentiment word from a post, as listed in the KNU lexicon. In previous studies (e.g., [66]), the researchers added the positive and negative sentiment scores together to infer the emotional state of the document writer from the total score. In contrast, we converted the negative sentiment scores into absolute values and used the positive and negative scores separately, based on a widely-held psychological understanding of the independence of positive and negative affect [67]. Iterative screening of the pandemic and endemic data yielded 1,740 sentiment scores each for positive and negative affect.

**Statistics.** We conducted the Mann-Whitney U test to further compare the levels of affect, as a non-parametric two-group comparison of means [68] due to the different sample sizes between the pandemic ($N = 1,132$) and endemic ($N = 608$) data, using IBM SPSS 27 statistical

software. We used means comparisons to test whether the levels of negative and positive affect expressed in the writings of depressed young South Koreans had changed significantly in the endemic condition.

## Results

### Descriptive information about the data

Fig 1 shows the number of endemic blues posts per month from May to December 2023. Below is an example of an endemic blues post translated into English:

> I am a student who was diagnosed with depression by the hospital. I am currently taking medication, but it does not seem to help me. My life is difficult because of my depression. I am skipping classes. It is so difficult. I feel too lethargic and depressed to do anything. I have given up my hobbies because I feel lazy. I wish I could attend school. My friends call me, but I do not want to attend school. I skip classes and kill time by looking at my smartphone in bed.

According to our manual coding of the sources of stress, contextual situations, and related thoughts and feelings, 199 posts were adolescent and 170 posts adult depressive episodes. We could not identify the age range of the writers of 239 posts.

### Corona blues topic models

Table 1 shows the pandemic blues topic model described in Kim [51] on the left and the endemic blues topic model used in this study on the right, along with the topic's high-frequency tokens for each topic and the topic proportion (%), which is the relative number of narratives a topic had in the data.

The pandemic blues topics (P1–P6) in Kim [51] are listed by the highest proportion of topics; however, the proportional differences between topics P1, P2, and P3 were small, and there was a similarity between topics in terms of thematic domains. Specifically, slightly more than half of the narratives (P2 and P4–6, 54.2%) were devoted to young South Koreans' questions and doubts about depression treatments. The remaining factors (P1 and P3) explained the risk factors for pandemic blues. Topic P1 (25.3%) described young South Koreans' difficulties in maintaining friendships (time, quality) due to life restrictions, while topic P3 (20.5%) concerned unstable family environments such as parental divorce, which contributed to the

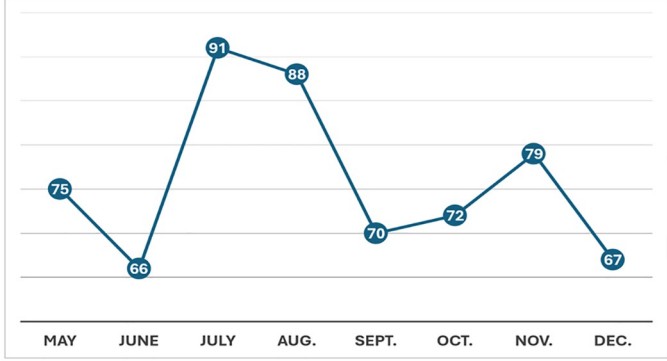

**Fig 1. Endemic blues posts per month.**

Table 1. A comparison between the pandemic and endemic blues topic models.

| Pandemic Blues | | | Endemic Blues | | |
|---|---|---|---|---|---|
| (February 2020–April 2023) | | | (May–December 2023) | | |
| Topic | Tokens | Prop. | Topic | Tokens | Prop. |
| P1. Friendship difficulties and their negative consequences | Friend, hospital, depressed, lethargy, time | 25.3% | E1. Seeking for depression test and treatment | Psychiatry, test, cure, survey, professional | 49.4% |
| P2. Symptoms caused by psychotropic medication | Symptom, hospital, medication, prescription, panic disorder | 23.5% | E2. Depressed adults' stress and barriers to treatment | Medical certificate, menopause, insomnia, medical bill, insurance | 18.01% |
| P3. Family-related depressive experiences | Mom, parents, emotions, counseling, mental | 20.5% | E3. Experiences of psycho-physical symptoms comorbid with depression | Panic disorder, bipolar disorder, anxiety disorder, concentration, weight | 9.63% |
| P4. Doubting the effectiveness of current treatment and seeking alternative treatments | Treatment, psychiatric_ department, medication, medicine, insomnia | 15.5% | E4. Thoughts about daily responsibilities | Week, interpersonal_ relationship, Monday, school life, responsibility | 8.32% |
| P5. Questions about psychotropic medication | Psychiatric_ department, headache, worry, morning, evening | 10.5% | E5. Depressed adolescents' family-related struggles | Minor, trauma, family history, Self-esteem, orphanage | 8.1% |
| P6. Depression-related insurance issues | Insurance, boyfriend, subscription, hospital, actual cost | 4.7% | E6. Social isolation due to depression and additional symptoms | Shame, self-hatred, laziness, COVID-19, Internet addiction | 6.54% |

Prop = topic proportion; pandemic blues results are those reported in Kim [51]

development of depression during the pandemic when people experienced stay-at-home orders and quarantine.

Compared with the pandemic blues topics, friendship maintenance issues (P1) diminished by the end of the pandemic. Young South Koreans' expressions of interest in professional psychological testing and treatment for depression (E1) continued to be the main topic with the highest proportion (49.4%). Topic E2 (18.01%) described age-specific conditions, such as menopause (and tokens about Korean men's military service, although not listed in Table 1) and concerns about the affordability of treatment for adults. Topic E3 (9.63%) showed that other psychological symptoms had developed under endemic blues conditions. Mood disorders (e.g., panic, bipolar, and anxiety disorders) and their cognitive and physical symptoms—such as poor concentration and weight changes—were the most frequently mentioned. Although not listed in Table 1, experiences of trauma and alcohol dependence were also evident in the results. Topic E4 (8.32%) described young South Koreans' thoughts and reflections on their daily responsibilities and duties, including their interpersonal relationships. Topic E5 is related to topic P3, which was identified during the pandemic. Although it decreased proportionally (from P3 20.5% to E5 8.1%), young South Koreans continued to experience stress resulting from damage to the family system. Finally, topic E6 showed longitudinal effects with the emergence of strong negative emotions such as shame and self-hatred and a tendency to self-isolate among those who were depressed during and after the pandemic. Although not listed in Table 1, "trouble," (poor) "diet," "paranoia," and "adjustment difficulties" were mentioned in an elaboration of the severity of the young South Koreans' situations.

In contrast to the pandemic blues topics, the endemic blues topics E2 and E5 showed age-specific domains of stress. The results of the subgroup (adolescents and adults) topic modeling are presented in Table 2.

In the adolescents' endemic blues topic model, a sizable proportion expressed interest in depression assessment and treatment (topic M1, 58.77%). However, being underage was a barrier to seeking professional help if they did not have the support or consent of their parents or

Table 2. A comparison between the adolescents' and adults' endemic blues topic models.

| Adolescents (*n* = 199) | | | Adults (*n* = 170) | | |
|---|---|---|---|---|---|
| Topic | Tokens | Prop. | Topic | Tokens | Prop. |
| M1. Minor's interest in depression test/diagnosis and treatment | Test, psychiatry, legal guardian, medical bill, medical referral | 58.77% | A1. Difficulties in daily functioning and interest in depression treatment | Psychiatry, trouble, lethargy, office worker, college student, COVID-19 | 26.2% |
| M2. Hospital treatment of underage students for depression | Medical certificate, school principal, pediatrician, university hospital, office of education | 14.17% | A2. Depression diagnosis, treatment, and military duty | Medical certificate, mental disorder, psychological test, military workforce, administration, medical records | 25.24% |
| M3. Comorbid symptoms of depression and their impact on daily life | Paranoia, memory, insomnia, interpersonal_ relationships, self-esteem | 12.89% | A3. Seclusion or isolation of adults who have experienced negative event(s) in the growth process | High school, middle school, lies, trauma, self-hatred | 17.94% |
| M4. Depression and eating problems | Gym, social phobia, diet, COVID-19, reflux esophagitis | 9.78% | A4. Periodic risks of depression and interest in treatment | Menopause, insomnia, lethargy, breast cancer, maternity leave | 17.03% |
| M5. Unstable family environment | Orphanage, family discord, unconsciousness, fear, box cutter | 4.39% | A5. Questions about medication for depression and co-morbidities | Alprazolam, side effects, cold sweat, Concerta, weight | 13.58% |

Prop = topic proportion

legal guardians. Topic M2 (14.17%) was the description of those who were officially referred by their school for hospital treatment. As shown in topics M3 (12.89%) and M4 (9.78%), depression and its comorbid symptoms led to difficulties in the adolescents' lives. The symptoms affected cognitive functions (e.g., memory and thinking skills, although not all tokens are listed in Table 2), but also affected weight changes and caused the adolescents' particular stress and concerns about interpersonal (peer) relationships. Mostly due to weight gain, signs of eating disorders appeared from the adolescents' frequent use of tokens such as gyms and diet, social, phobia, and reflux esophagitis as by-products of problematic eating (and other tokens included self-hatred, complex, junk, and laziness, although not listed in Table 2). We retained Topic E5 in Table 1 (depressed adolescents' family-related struggles) as a separate topic for adolescents (M5, 4.39%) with an additional sign of risk of self-injury (token "box cutter").

Among adults, interest in seeking treatment was the most common (A1, 26.2%), along with references to difficulties in fulfilling adult responsibilities and obligations at school and work. The age-specific characteristics of adults were shown by frequent questions and expressions of frustration about military service (A2, 25.24%) and periodic susceptibility to depression, especially in women during menopause, during and after a battle with breast cancer, and after giving birth (A4, 17.03%). Congruent with topic E6 in Table 1 (social isolation due to depression and additional symptoms), the self-isolating tendency of depressed adults was observed (A3, 17.94%), elaborating on the reasons for social withdrawal stemming from their past experiences (although not listed in Table 2, additional tokens included repeating university entrance exams, and motivation). Finally, topic A5 (13.58%) showed that numerous adults had knowledge of psychotropic medications for depression and comorbid symptoms of depression, such as loss of attention. References to specific drug names, such as Alprazolam and Concerta, were found with descriptions of perceived side effects such as cold sweats, weight changes, dizziness, and dependence, although not all these tokens are listed in Table 2.

## Topic changes in the COVID-19 endemic era

In the results of the DMR analysis, changes in the topic proportion estimates explain how the narrative focus of young South Koreans shifted from the endemic declaration to the end of 2023. We drew the trend graphs at two-month intervals to improve readability. As shown in

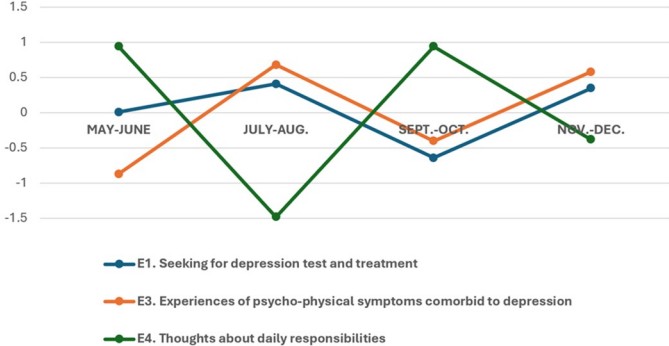

**Fig 2. A trend in discussions about the endemic blues and daily responsibilities.** Y-axis = The degree of proportional change (increase or decrease) of a topic compared to other topics within the data at the given time.

Fig 2, the comparative number of narratives focusing on topics E1 and E3 showed an inverse trend to topic E4. Young South Koreans discussed their thoughts about their daily responsibilities and duties soon after the end of COVID-19. Meanwhile, conversations about psychological struggles and treatment needs diminished but soon resumed. This pattern then recurred.

Fig 3 describes how young South Koreans' focus on each domain of distress changed over time. In terms of age, adults' struggles with additional factors of the endemic blues, such as periodic risks and obstacles to treatment (E2), seemed to cease after fall; however, an increasing tendency to self-isolate (E6), which was found predominantly among adults (Table 2, A3), showed a prolongation of adversity. Adolescents' struggles at home (E5) showed a decreasing trend for several months but increased towards the end of the year.

## Comparison of pandemic and endemic affect

The mean comparison results between the levels of positive and negative affect (examined separately) shown in young South Koreans' narratives about the corona blues during the pandemic and since the endemic declaration are presented in Table 3. Neither the positive nor negative affect changed in a statistically significant manner ($p > .05$) despite the end of COVID-19 and the restoration of resources.

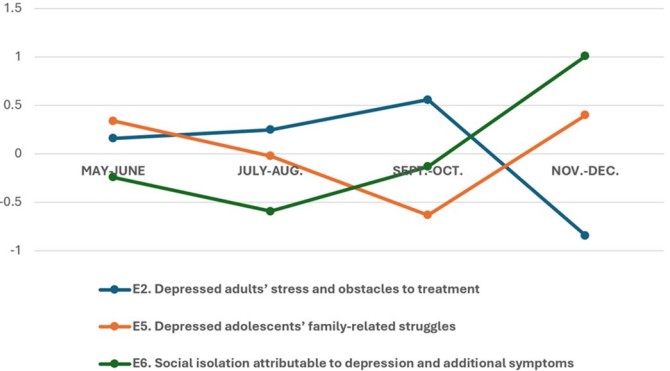

**Fig 3. A trend in the discussion of additional factors and by-products of depression.** Y-axis = The degree of proportional change (increase or decrease) of a topic compared to other topics within the data at the given time.

**Table 3. Comparison of pandemic and endemic mean affect scores.**

|  | Group (N) | Mean | SD | Mean Rank | U | z | p |
|---|---|---|---|---|---|---|---|
| Positive affect | Pandemic (1132) | 4.53 | 5.86 | 888.06 | 326509.00 | -1.78 | .076 |
|  | Endemic (608) | 4.26 | 6.23 | 841.52 |  |  |  |
| Negative affect | Pandemic (1132) | 25.06 | 19.75 | 872.22 | 342184.00 | -.20 | .846 |
|  | Endemic (608) | 25.45 | 21.04 | 867.30 |  |  |  |

*SD* = standard deviation

## Discussion

Despite the shift of the COVID-19 pandemic to endemic status, its psychological aftermath has been widespread worldwide. In particular, the corona blues aspects of fear, lethargy, and loneliness continue to affect young people [17, 69]. Various private and social welfare services, including face-to-face and distant/virtual counseling, have been provided to support the mental recovery of the public, but young people experience uncertainty about the future and low self-efficacy. However, they have little energy to overcome these due to the endemic blues [70]. Mental health professionals have expressed concern about longitudinal effects, such as increased suicide rates, reflecting the precedents of the SARS and MERS outbreaks [16–18]. Evidence from past epidemics suggests that restoring resources for education, job (search), and leisure may not be sufficient to stop the endemic blues [14, 15].

To explore the state of endemic blues among young South Koreans, we used 1,740 online posts on depression shared during the pandemic and after the status was changed to endemic. Our findings reveal emerging topics of depression since the pandemic and changes in narrative focus as young South Koreans grapple with the lingering effects of COVID-19. Beyond this exploration, we compared the levels of affect expressed during the pandemic and since the declaration of endemic to examine whether the negative (and positive) affect remained statistically unchanged, as found in follow-up studies after SARS and MERS [19, 20].

### Is the residual influence of COVID-19 present in a group of depression-related online posts shared after the pandemic?

Topic modeling results showed that several pandemic blues topics remained in the endemic blues topic model, and direct references to "COVID-19" were found in an emerging endemic blues topic. Treatment-seeking by young South Koreans continued during and after the pandemic phase in a different way. Questions and expressions of doubt have evolved into an interest in finding resources for professional psychological assessments and treatments. Previous studies [71, 72] have also anticipated that the provision of self-help information and virtual resources (e.g., self-management guidelines and videos) as a mental health crisis intervention measure during a pandemic could lead to the public gaining knowledge about mental health and reducing resistance to treatment.

The residual influence of COVID-19 was evident in experiences of extended symptoms (e.g., panic disorder and weight gain) and signs of additional adversity of long-term depression (e.g., self-isolation tendencies). There were realistic barriers to treatment (e.g., cost and insurance). Recent reports of nationwide statistics on the quality of life of young South Koreans [73, 74] are consistent with the increasing numbers of socially isolated young people (3.1% before COVID-19 and 5% in 2021) with poor mental health. Alarmingly, 75.4% of the socially isolated respondents (*N* = 8,436) reported suicidal thoughts [75].

## What are the main themes (causes and content) of the endemic blues?

Endemic blues topics can be divided into the following domains: a) interest in treatment (Table 1, E1), b) factors that maintain depression (E2, E5), c) by-products of depression (E3, E6), and d) concerns about life (E4). Adolescents' interest in seeking treatment was often interrupted by difficulties in obtaining support from parents or legal guardians unless they were referred for treatment by their school. In all studies (e.g., [76]), family instability and lack of communication made it difficult for psychologically vulnerable adolescents to seek help. A wide range of family factors—such as parental psychiatric history, parenting style, and family climate—are strongly related to the development of depression and treatment engagement [77, 78].

In contrast, adults' interest in treatment stems from their experience of maladaptation to normal life. Comparable results were found in studies of people with post-COVID syndrome, including depressive symptoms [79, 80]. In addition to neurobiological responses, this syndrome causes fatigue, sleep disturbance, and cognitive dysfunction in COVID-19 survivors. Another stream was a country-specific topic found among South Korean men subject to military service. They sought official means to receive psychological assessments for depression as the Korean army strengthened new administrations when COVID-19 was declared endemic. A previous text analysis [81] showed consistent results when exploring young South Koreans' concerns about self-harm.

In terms of perpetuating factors, depressed adolescents related their ongoing struggles to family factors—indicating a risk of self-harm—whereas adults mentioned periodic and event-specific sources of negative mood, such as menopause and experiences during and after treatment for breast cancer and childbirth. Our findings on these additional factors have been highlighted in previous studies on groups at risk of depression [82] adolescents in unstable families [83], climacteric women [84], breast cancer survivors, and [85] postpartum depression. These results suggest that situational risks may interact with the residual influence of COVID-19 on the maintenance of endemic blues.

In terms of the by-products of depression, adolescents reported cognitive dysfunction, as found in other post-COVID studies of adolescent adjustment [86, 87]. In particular, they struggled with problematic eating (mostly binge eating) that began during the pandemic and expressed concerns about their appearance and complications, such as reflux esophagitis. This is consistent with findings during the pandemic (e.g., [88]) on the relationship between COVID-19, eating, and physical (in)activity. The COVID-driven negative emotions, including depression, led to increased emotional eating and obesity as psychosomatic outcomes in all age groups, especially in juvenile females [89, 90]. In contrast, adults who remained depressed tended to withdraw and isolate themselves from society, disregarding opportunities for social interaction. Previous studies on the maintenance of self-isolation (i.e., self-quarantine) in the COVID-19 era [91, 92] suggest that long-term isolation could weaken the social support systems of vulnerable people with low coping skills.

## How have the themes changed over time since the declaration of COVID-19 as endemic?

From the time of the endemic declaration until the end of 2023, for about six months, changes in the narrative focus of young South Koreans showed that they put their need for treatment aside when distressing thoughts about daily responsibilities arose; however, the remaining depression and comorbid symptoms of depression brought their attention back to seeking treatment. This recursive pattern suggests that South Koreans struggle to find easy access to mental health resources while remaining on track for education and work. In

this regard, recent national reports have shown that less than a majority (44.5%, $N = 11,819$) of South Korean schools (elementary to high schools) have professional school counselors [93], and that there are a below-average numbers of psychiatrists and psychologists working in the medical field (.08 per 1,000 people in South Korea, $M = .18$ per 1,000 people) compared to other OECD countries [94]. Meanwhile, depressed South Korean adults' discussions of additional factors contributing to the persistence of depression and realistic barriers to treatment gradually increased but decreased towards the end of the year. With these changes, descriptions of reclusive lifestyles have increased. These results indicate the long-term consequences of young people's social withdrawal. Previous Studies on depression (e.g., [95]) showed that weak social connectedness had a stronger association with clinical depression than other factors such as age, gender, and employment status. Our findings support this, showing that psychological sequelae are likely to persist, resulting in social costs.

## Has the level of negative affect decreased statistically significantly during the endemic compared to the pandemic era?

The statistically indifferent emotional state that persists despite the "return to normal" also shows that the endemic blues and its by-products may not spontaneously subside in a few years, given the length of time people remained under the influence of COVID-19 and the extent of life disruption compared to the SARS and MERS outbreaks [21]. Similar concerns have been raised that the psychological sequelae may have delayed effects, particularly on self-harm [96] and suicide [97], if not addressed by timely interventions. Measures to reduce hopelessness and practice adaptive coping skills are needed to help young people become resilient [98].

## Strengths and limitations

This study used a multidisciplinary approach to explore the existence and content of endemic blues and to compare the findings with those found during the pandemic. Seminal reviews (e.g., [11]) and meta-analyses (e.g., [2]) have summarized the psychological struggles of the public during COVID-19. However, few studies have compared the states of corona blues during the pandemic, and after the change of status to endemic, despite calls for this study [99] and concerns about the residual effects of COVID-19 [14]. Based on these needs, the strength of this study lies in the comparisons between the corona blues related topics that emerged from young people's autobiographical writings at different points in time. Text mining of online posts allowed us to collect naturalistic narratives free of time and space constraints, and concerns about social desirability or observer effects. Together, the results of the text-mined sentiment analysis and mean comparisons were valuable in demonstrating the existence of psychological sequelae.

Simultaneously, this study has limitations that should be addressed in future studies. First, the anonymous online post data were not identifiable in terms of the writers' genders or whether individuals had been clinically diagnosed with depression. Posts were only included if pathological keywords such as "depress(ion)," "depressed," "depressi(on/ve) symptoms," or "major depressive disorder" were found in the body of the text and tokens extracted from the topic modeling process helped to infer writers' demographic information such as their age range and gender; however, specific studies of meaningful samples are needed. For example, depression topics relevant to a clinical sample may differ from those of mild depression. Some topics may be more salient for women than men, and vice versa. Relatedly, we cannot ascertain whether all posts were written by different authors. Following the means used in previous text-

mining studies to avoid duplicates [51, 52], we excluded posts that were shared on the same day with identical post titles and body content. However, some authors may have intentionally posted more than once. Given the size of the data (a total of 1,740 posts), it is unlikely that the group of frequent writers would have had a predominant influence on our data, but this could still be a confounding factor. Future studies could include surveys and text mining methods to control for the confounder related to anonymity.

Second, some results showed country- and culture-specific characteristics of young South Koreans that could not be confirmed within the context of this study. The comparison of the pandemic and endemic contexts through topic models (i.e., from questions and doubts to interest in professional treatment) and the importance of a change trend in the endemic era (i.e., difficulty in maintaining thoughts about daily responsibilities and seeking treatment simultaneously) may reflect the lack of resources and opportunities for mental health care. In the long term, the self-isolation tendency of depressed young adults in South Korea—where people adhere to collectivistic cultural values—may lead to greater adversity than in individualistic cultures. The military service of depressed young South Korean men in the endemic era is also worthy of further study. Finally, this study focused on psychological sequelae, and mainly on the corona blues as a type of residual problem more than three years after the pandemic began. However, other factors such as the economic recession, unemployment, and maladjustment to the "new normal" should be considered to understand the current state of young people holistically.

## Conclusions

Our findings reveal the residual effects of COVID-19 on young South Koreans' depressive psychology through text mining of online posts shared during the pandemic and after the change of status to endemic. Topic models describing endemic blues revealed that young South Koreans' interest in professional depression assessment and treatment is high; however, there are some realistic barriers, mostly due to difficulties in finding easily accessible treatments while fulfilling daily responsibilities and duties. Age-specific topics described various additional factors (e.g., unstable family factors for adolescents and menopause or military service for adults) and by-products of prolonged endemic blues by age group (e.g., eating problems for adolescents and self-isolating tendencies for adults). Over time, the changes in the number of narratives focusing on different topics in the endemic era implied that timely assistance to help the depressed find accessible treatment is critical to prevent long-term adversities such as chronically depressed young people's withdrawal from society and suicidal behavior.

## Author Contributions

**Conceptualization:** Seoyoung Kim, TaeYoon Aum, Dong-gwi Lee.

**Data curation:** TaeYoon Aum.

**Formal analysis:** Seoyoung Kim.

**Methodology:** Seoyoung Kim.

**Project administration:** Seoyoung Kim, Dong-gwi Lee.

**Software:** Seoyoung Kim.

**Supervision:** Dong-gwi Lee.

**Validation:** TaeYoon Aum, Dong-gwi Lee.

Writing – **original draft:** Seoyoung Kim.

Writing – **review & editing:** TaeYoon Aum, Dong-gwi Lee.

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
