## [Decision Letter · Decision Letter 0]

20 Aug 2024

PONE-D-24-16610Depression in the COVID-19 endemic era: Analysis of online self-disclosures by young South KoreansPLOS ONE

Dear Dr. Lee,

Thank you for submitting your manuscript to PLOS ONE. After careful consideration, we feel that it has merit but does not fully meet PLOS ONE’s publication criteria as it currently stands. Therefore, we invite you to submit a revised version of the manuscript that addresses the points raised during the review process.

We look forward to receiving your revised manuscript.

Kind regards,

Michal Ptaszynski, PhD

Academic Editor

PLOS ONE

Journal Requirements:

3. In the online submission form, you indicated that [Data (i.e., raw autobiographical online posts about depressive experiences written in Korean) cannot be shared publicly because of a possibility of misuse or misunderstanding about Korean language. Data are available from the first author upon a reasonable request sent to kimalexis1129@gmail.com.]. 

Reviewers' comments:

Reviewer's Responses to Questions

**Comments to the Author**

1. Is the manuscript technically sound, and do the data support the conclusions?

Reviewer #1: Yes

Reviewer #2: Yes

2. Has the statistical analysis been performed appropriately and rigorously? 

Reviewer #1: Yes

Reviewer #2: Yes

3. Have the authors made all data underlying the findings in their manuscript fully available?

Reviewer #1: No

Reviewer #2: Yes

4. Is the manuscript presented in an intelligible fashion and written in standard English?

Reviewer #1: Yes

Reviewer #2: Yes

5. Review Comments to the Author

Reviewer #1: This is a well-written paper on an interesting topic. Specific comments are found below.

General comments

•An ethics statement should be provided, given that the study involved human subjects

Introduction

•Can be shortened

•Explain more clearly if there were previous studies looking at the endemic blues using text-mining and what did they find

Methods

•Please clarify the dates used for the endemic data analysis

•Please clarify if the pandemic data and endemic data were analyzed from the same online portal

•Please explain how the endemic posts were categorized into adult/adolescence/unknown. Were the pandemic posts also categorized into adult/adolescence using the same methods?

•Are the posts all from different authors? Namely, one post per author? If not, how do the authors deal with this possible confound (e.g., many posts can be from one author, making the pool not representative of the population)? Please explain in the Methods.

•The Authors can add a “Statistics” section at the end of the Methods and include there all statistical tests conducted

Results

•Please add standard errors (STE) to Figures 2,3.

•Figure 3: is E6 across adults and adolescents?

Reviewer #2: Thank you all the authors for this interesting study. Overall, the study presents a strong contribution and understanding of the enduring impact of COVID-19 on mental health focusing on young South Koreans but would benefit from addressing these feedbacks to enhance clarity and practical relevance.

Clarity

- The manuscript addresses an important and timely topic: the residual effects of COVID-19 on mental health, with a specific focus on young South Koreans during the transition from pandemic to endemic phases. The relevance of this research is clear, given the global mental health crisis exacerbated by the pandemic.

- While the study aims to compare the effects of the pandemic and endemic periods, a more detailed articulation of the specific hypotheses being tested would provide a clearer focus and enhance the study’s impact. Additionally, the manuscript should better justify how the findings might generalize beyond the South Korean context.

Methodology

- The use of LDA and DMR methods reflects a robust understanding of text mining techniques. The selection of six topics for the LDA model is logical. However, the manuscript should provide a more detailed justification for choosing five topics for subgroup analyses. A discussion on how the number of topics balances interpretability with detail would be beneficial.

- Metadata in DMR: Including publication month as metadata in the DMR analysis is a strength as it facilitates tracking changes over time. Are there any potential limitations or biases introduced by this approach?

Results and discussion

- The findings highlighting the persistence of symptoms like panic disorders and weight gain are significant. The manuscript should further explore the implications of these residual effects and discuss how they might inform future mental health interventions.

- The observed changes in narrative focus over time are intriguing. The manuscript should provide a deeper analysis of potential reasons for these changes and consider how external factors, such as policy shifts or societal changes, might influence these trends.

- The lack of statistically significant changes in affect levels between the pandemic and endemic periods is notable. The manuscript should discuss possible reasons for this finding, including whether the methods used were sensitive enough to detect subtle changes or if other external factors may have influenced the results.

- The inability to identify author demographics or clinical diagnoses is a significant limitation. The manuscript should acknowledge this and propose how future research might address this issue.

- The focus on South Korea may limit the generalizability of the findings. The manuscript should discuss how culturally specific factors might influence the results and how these findings might apply to other contexts.

- The paper mentions cultural factors but does not fully explore how these might influence the results. A more detailed examination of cultural influences and comparisons with other countries could enhance the discussion.

- The conclusions could be strengthened by offering more actionable recommendations for policymakers and mental health professionals. Specific suggestions on how to address the identified issues could provide practical value.

6. PLOS authors have the option to publish the peer review history of their article (what does this mean?). If published, this will include your full peer review and any attached files.

Reviewer #1: No

Reviewer #2: No

---

## [Author Response · Author response to Decision Letter 0]

3 Nov 2024

Comment 1: An ethics statement should be provided, given that the study involved human subjects

Response 1: Thank you for your comment on this important point. Our study did not involve human subjects. Instead, we used anonymous online posts/articles that were voluntarily and publicly shared on the Knowledge-In forum (a South Korean web forum: https://kin.naver.com/). This type of text-mining study does not require IRB approval because no procedure involves direct contact with participants. We have added an explanation on this matter on page 8 (lines 157-159) of our revised manuscript with track changes. 

Comment 2: Introduction can be shortened

Response 2: Following your suggestion, we have shortened the Introduction section by removing non-essential information and examples. 

Comment 3: Explain more clearly if there were previous studies looking at the endemic blues using text-mining and what did they find.

Response 3: Although we could not find a previous text-mining study that focused specifically on the “endemic” blues, there were a few text-mining studies on the Corona blues. We elaborated on the previous results on page 7 (lines 124-130 of our revised manuscript with track changes). The results showed not only that COVID-19 caused an emotional crisis to the public, but also that a collection of online texts is useful phenomenological data.

Comment 4: Please clarify the dates used for the endemic data analysis

Response 4: The topic modeling data analysis was performed on March 26, 2024 (data collection: January 14, 2024). We specified each date on page 8 (line 162) and 10 (line 203) of our revised manuscript with track changes.

Comment 5: Please clarify if the pandemic data and endemic data were analyzed from the same online portal

Response 5: Yes, both data were collected/scraped from the Knowledge-In forum of the Naver portal. This explanation has now been added to page 8 (lines 162-163 of our revised manuscript with track changes). Since we obtained the pandemic data from Bum-Hyun Kim, we used the same search terms and inclusion criteria that Kim used to collect our endemic data in order to make a fair comparison between the pandemic and endemic blues topic models.

Comment 6: Please explain how the endemic posts were categorized into adult/adolescence/unknown. Were the pandemic posts also categorized into adult/adolescence using the same methods?

Response 6: Regarding the method we used to categorize the endemic posts into “adult”, “adolescent”, and “unknown”, we have added detailed explanations on page 10 (lines 187-196). At the same time, we acknowledge that the method is not unambiguous because the posts are from anonymous authors. This limitation is noted on page 25 (lines 487-492 of our revised manuscript with track changes). The pandemic posts were not categorized, because our study focus was on the endemic blues. We used the pandemic posts to compare them with the endemic posts and to confirm the existence of psychological sequelae by finding topics about depression-related struggles that had persisted into the endemic era. The purpose of the categorization (for subgroup analysis within the endemic data) is now specified on page 8 (lines 146-148) to clear up the confusion.

Comment 7: Are the posts all from different authors? Namely, one post per author? If not, how do the authors deal with this possible confound (e.g., many posts can be from one author, making the pool not representative of the population)? Please explain in the Methods.

Response 7: Thank you for pointing out about the potential confounder. In an effort to control the confounder, following the means used in previous text-mining studies, we case-wise deleted all posts shared on the same day with identical title and body content to eliminate duplicates (explained on page 9, line 170 of our revised manuscript with track changes). However, we acknowledge that there is still a remaining possibility that some posts are from the same author(s). This is added as a limitation on page 26 (lines 495-503).

Comment 8: The Authors can add a “Statistics” section at the end of the Methods and include there all statistical tests conducted

Response 8: Following your suggestion, the “Statistics” section is now added on page 12 (line 250 of our revised manuscript with track changes). The only psychological statistical method used in our study was the Mann-Whitney U test for a comparison of means between the levels of affect/sentiment expressed in the pandemic and endemic data. 

Comment 9: Please add standard errors (STE) to Figures 2,3.

Response 9: The topic proportion estimates, or proportional changes, do not have the STE, unlike the values obtained from traditional psychological statistics. The proportional changes shown on the y-axes of Figures 2 and 3 describe the degree of variation in the topic proportion estimates at a given point in time. To resolve this confusion, we have added the definition of the y-axis to the captions of Figures 2 and 3 on page 19 (lines 349-350 and 359-360 of our revised manuscript with track changes). A growth indicates that the topic was discussed more passionately than other topics by depressed young South Koreans at the time. The topic may decline at a later point in time as trends change. The score itself has no meaning, but the variation can reveal the trend changes.

Comment 10: In Figure 3, is E6 across adults and adolescents?

Reseponse 10: In Figure 3, we added “predominantly among adults” next to the E6 “Social isolation due to depression and additional symptoms”, based on the topic’s tokens and a review of the raw data/original posts (as described on pages 17-18, lines 321-325 of our revised manuscript with track changes).

---

## [Decision Letter · Decision Letter 1]

19 Nov 2024

Depression in the COVID-19 endemic era: Analysis of online self-disclosures by young South Koreans

PONE-D-24-16610R1

Dear Dr. Lee,

We’re pleased to inform you that your manuscript has been judged scientifically suitable for publication and will be formally accepted for publication once it meets all outstanding technical requirements.

Kind regards,

Michal Ptaszynski, PhD

Academic Editor

PLOS ONE

Additional Editor Comments (optional):

Reviewers' comments:

Reviewer's Responses to Questions

**Comments to the Author**

1. If the authors have adequately addressed your comments raised in a previous round of review and you feel that this manuscript is now acceptable for publication, you may indicate that here to bypass the “Comments to the Author” section, enter your conflict of interest statement in the “Confidential to Editor” section, and submit your "Accept" recommendation.

Reviewer #1: All comments have been addressed

Reviewer #2: All comments have been addressed

2. Is the manuscript technically sound, and do the data support the conclusions?

Reviewer #1: Yes

Reviewer #2: Yes

3. Has the statistical analysis been performed appropriately and rigorously? 

Reviewer #1: Yes

Reviewer #2: Yes

4. Have the authors made all data underlying the findings in their manuscript fully available?

Reviewer #1: (No Response)

Reviewer #2: Yes

5. Is the manuscript presented in an intelligible fashion and written in standard English?

Reviewer #1: Yes

Reviewer #2: Yes

6. Review Comments to the Author

Reviewer #1: (No Response)

Reviewer #2: Thank you all the authors for this interesting study. The study presents a strong contribution and understanding of the enduring impact of COVID-19 on mental health focusing on young South Koreans.

All of my commensts has been addressed. Thank you.

7. PLOS authors have the option to publish the peer review history of their article (what does this mean?). If published, this will include your full peer review and any attached files.

Reviewer #1: No

Reviewer #2: No

---

## [Editor Report · Acceptance letter]

12 Dec 2024

PONE-D-24-16610R1 

PLOS ONE

Dear Dr. Lee, 

I'm pleased to inform you that your manuscript has been deemed suitable for publication in PLOS ONE. Congratulations! Your manuscript is now being handed over to our production team.

Kind regards, 

on behalf of

Dr. Michal Ptaszynski 

Academic Editor

PLOS ONE